# AdaTune: Adaptive Tensor Program Compilation Made Efficient

**Menghao Li**[*] **Minjia Zhang**[*] **Chi Wang** **Mingqin Li**
Microsoft Corporation
{t-meli,minjiaz,wang.chi,mingqli}@microsoft.com

## Abstract

Deep learning models are computationally intense, and implementations often have to be highly optimized by experts or hardware vendors to be usable in practice. The DL compiler, together with *Learning-to-Compile* has proven to be a powerful technique for optimizing tensor programs. However, a limitation of this approach is that it still suffers from unbearably long overall optimization time. In this paper, we present a new method, called AdaTune, that significantly reduces the optimization time of tensor programs for high-performance deep learning inference. In particular, we propose an adaptive evaluation method that statistically early terminates a costly hardware measurement without losing much accuracy. We further devise a surrogate model with uncertainty quantification that allows the optimization to adapt to hardware and model heterogeneity better. Finally, we introduce a contextual optimizer that provides adaptive control of the exploration and exploitation to improve the transformation space searching effectiveness. We evaluate and compare the levels of optimization obtained by AutoTVM, a state-of-the-art Learning-to-Compile technique on top of TVM, and AdaTune. The experiment results show that AdaTune obtains up to 115% higher GFLOPS than the baseline under the same optimization time budget. Furthermore, AdaTune provides 1.3–3.9× speedup in optimization time over the baseline to reach the same optimization quality for a range of models across different hardware architectures.

## 1 Introduction

The enormous computational intensity of Deep Neural Network (DNN) models has attracted great interest in optimizing their performance. In particular, popular deep learning (DL) frameworks such as TensorFlow [6] and PyTorch [32] adopt custom optimized kernels such as Nvidia cuDNN [15] or Intel MKL-DNN [2] as back-end. However, given the increasing complexity of tensor operations in DNNs and the volatility of DL algorithms, it calls for developing fast and automated compilation frameworks to handle the unprecedented amount of innovations. To imitate or even surpass the success of hand-optimized libraries, recent research has developed neural network compilers, such as XLA [4], Halide [36], Glow [37], Tensor Comprehension [40], and TVM [13]. Among them, TVM has shown superior performance improvements using a technique called *Learning-to-Compile* (AutoTVM) [14]. AutoTVM optimizes the code by generating many versions of a tensor program and chooses the best through simulated annealing search over a large space of code transformation choices. Furthermore, it employs a learned cost model trained by actual hardware performance measures to predict the performance of diverse inference computations on real hardware targets.

While the *Learning-to-Compile* approach produces highly optimized code of DNN models, they may suffer from excessively long optimization time. As an example, although AutoTVM is able to demonstrate close to 2× performance improvement over TensorFlow on ResNet-18, the compilation time can still take several hours or even tens of hours [14]. With the active research that has been pushing the model size to millions or even billion-scale parameters with a training time of only a

---

[*]Both authors contributed equally. Order of appearance is random.

few hours or less than one hour [46, 19, 47, 29, 38, 48], it makes reducing the DL compilation time for inference of the current solution even more prominent. Furthermore, since many of these DL compilers have been adopted by major players in the industry [43, 44, 26, 30], many users of these pipelines, including deployment engineers, would have to go through the optimization numerous times. Finally, as new neural architectures [18, 41, 16, 34] come out in rapid speed, and with deeper or wider networks [45, 38, 35, 5] on various hardware platforms [3, 22, 28], we are forced to optimize the networks more frequently. The excessive long optimization time hinders the turnaround time and even puts the practical utility of the current compiler-based solutions into question.

We aim at accelerating innovations by developing an automatic and efficient optimization process for DNN models. For this purpose, we introduce AdaTune, a method that achieves similar or better optimization quality but with shorter optimization time. Furthermore, AdaTune improves the adaptivity of LTC and reduces the hyperparameter tuning required, accelerating the productivity and agility of DNN model deployment. Specifically, the contributions of our paper consist of (1) a preliminary analysis that reveals the inefficiency and challenges of the existing approaches, (2) an *adaptive evaluator* that statistically determines the number of runs for performance measurement, (3) a *surrogate modeling with uncertainty quantification* that allows to better capture hardware and model heterogeneity, and (4) a *contextual optimizer* that provides control of exploration and exploitation dynamically to improve the transformation space searching effectiveness. We conduct extensive experiments to show that the proposed approach consistently outperforms the previous method on various models and hardware. It not only allows us to optimize DNN models 1.3-3.9× faster than the baseline to reach the same optimization quality but also obtains up to 115% higher GFLOPS under the same time budget. We conduct ablation analysis to study the effects of the proposed techniques, and we will make the source code publicly accessible to encourage further research.

## 2 Background and Related Work

**DL compilation pipeline.** DL compilers like TVM [13] have recently become popular for auto-matically optimizing DL programs [37, 4, 36, 40]. A typical DL compiler contains multiple passes to optimize a model trained by popular DL frameworks such as TensorFlow [6], PyTorch [32], or MXNET [12], as shown in Fig. 1. In the first pass (box with dotted line), the compiler frontend applies target-independent and white-box target-dependent optimizations that do not include a measure of actual execution time. The target-independent passes perform optimizations such as operator fusion and data layout transformation and the while-box target-dependent optimizations apply heuristic rules for code transformation based on domain knowledge, both of which do not need a specification of the target hardware. Recent work such as AutoTVM [14] extends the pipeline with another pass, which is a black-box target-dependent pass, which uses learning machinery to perform optimizations.

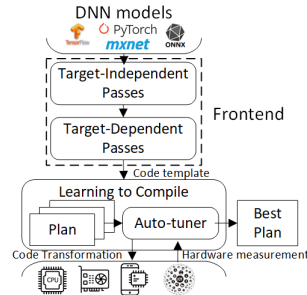

Figure 1: DL compilation pipeline.

Table 1: Example of TVM knobs.

| Knobs | Definition | Example values |
|---|---|---|
| tile_f | Loop tile decisions on the | 84 |
| tile_y | number of filters, and height | 140 |
| tile_x | and weight of feature maps | 140 |
| tile_rc | Loop tile reduction decision on | 2 |
| tile_ry | the number of channels, height | 2 |
| tile_rx | and weight of filters | 2 |
| auto_unroll max_step | The threshold of iterations a loop to be unrolled | 3 |
| unroll_explicit | Explicitly unroll loop | 2 |

**Black-box target-dependent pass.** In this pass, the compiler converts code transformation deci-sions as code templates, and it makes use of an auto-tuner (with optimization algorithm) and real hardware measurements to efficiently find the best transformation on target hardware (e.g., CPU, GPU, ARM, or IoT devices). Table 1 lists the knobs for code transformation of a convolution block on GPUs as example , which control various aspects of the optimization and determine whether the code (1) fully utilizes the internal parallelism within processors, (2) uses the shared memory wisely, and (3) maximizes data locality. Auto-tuning has been studied for generic program compilation

using domain-specific search techniques [9, 10, 42, 7]. Later, [14] builds on top of prior work by using a cost model and simulated annealing to search the transformation space for DNN models. However, there are certain limitations, as described in Section 3. Subsequently, [8] proposes to use reinforcement learning for efficient search space exploration. However, their approach involves a non-trivial amount of additional hyperparameter tuning as well as additional domain-knowledge on the validity of potential solutions on specific hardware.

**Problem formulation.** Given the vastness of the transformation space, if we denote the space as $S_e$ and function $Perf(\cdot)$ as the performance (i.e., GFLOPS – Giga floating point operations per second) of one transformation plan $p$ on a given hardware target $h$, the goal of the black-box target-dependent pass is to find a transformation plan $p*$ in $S_e$ that maximizes $Perf(\cdot)$ on $h$ over $S_e$ efficiently.

## 3 Preliminary Analysis

This section presents several studies that guided the design of the approach introduced in Section 4.

**Observation 1. DL compiler generates highly optimized code but results in prolonged optimization time.** Prior work [13, 14] uses an auto-tuner to select a plan $p$ from the transformation space, calls a code generator as a subroutine to generate machine code for that plan on a specific hardware, and then executes the generated code on the target hardware $n$ times to obtain an estimated cost of $p$ as $avg(p) = \frac{1}{n}\sum_{i=1}^{n} Perf(p, q_i)$, where $q_i$ represents an input of inference. Existing methods require the number of repeats $n$ as an input. However, for a fixed $n$, it is not clear a priori whether we should (a) run many repeats (large $n$) with more accurate measurement but also large measurement cost; or (b) consider a small number of repeats (small $n$) with inaccurate measurement but small measurement cost. $n$ cannot be too small, because then even if a transformation plan with low *average* cost is chosen, its *variance*, $std(p) = \sqrt{\frac{1}{n}\sum_{i=1}^{n}(Perf(p, q_i) - avg(p))^2}$ can be high. This is undesirable because then the measured average cost could deviate from its true performance due to the variance in the measurement, and we may end up choosing a suboptimal plan. Therefore, the general practice is to use a large conservative number for $n$ to achieve accurate estimates of the true cost, which is why it takes a very long time for existing auto-tuning methods to find a good solution. Figure 2 shows the execution time (on CPU) breakdown on 12 tasks from ResNet-18 [20]. As can be seen, the majority of time is spent on hardware measurement. Therefore, it is important to cut down the expensive cost of hardware measurements that examine the goodness of a plan.

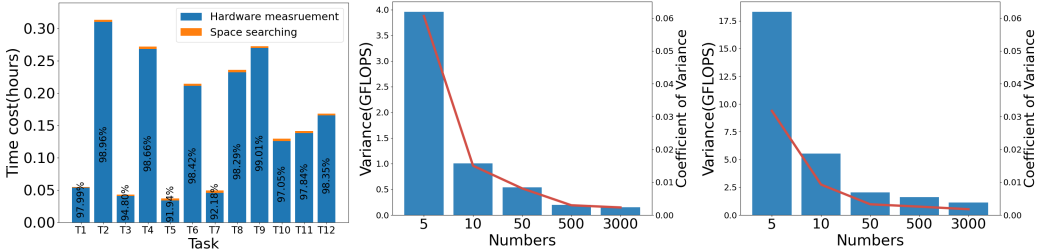

(a) Intel Xeon x86 CPU E5-2690 v3          (b) Nvidia Tesla P100

Figure 2: Performance breakdown of AutoTVM on ResNet-18.

Figure 3: Performance measurement variance and coefficient-of-variation on different hardware.

**Observation 2. Existing methods perform measurement without considering model diversity and hardware heterogeneity.** Prior study [14] claims that considering variance does not help. Therefore, they use a regression model (e.g., XGBoost) to learn and predict the performance of a transformation plan. Different from their observations, we find that in practice, based on the scenario, a model might need to be optimized against different hardware, including x86 CPU [43, 49], GPUs [21], ARM, and various ML accelerators [31, 25], all of which have very different architectures (e.g., memory hierarchy, instruction level parallelism, hardware prefetching, branch prediction, etc), which appear to have very different variance behaviors. Figure 3a and Figure 3b show the mean-variance range under x86 CPU and GPU for different $n$. As $n$ increases, the variance (bar) decreases

as expected. Although the absolute variance on GPU is higher than that on CPU, the coefficient of variation (curve) on CPU is much larger than that on GPU, especially when $n$ is relatively small. Furthermore, different models may also have diverse execution patterns, e.g., small/large model, regular/irregular computations, inter-op/intra-op parallelism, data dependencies, all of which can affect the run-to-run variance. Since existing work assumes the uncertainty is low by using a large $n$, if we reduce $n$, it is more likely that a regression model without accounting for the uncertainty can lead to suboptimal trials.

**Observation 3. Existing approaches employ static exploration-vs-exploitation, which limits the adaptivity of the compiler optimization.** Existing works [13, 14] employ the genetic algorithm or simulated-annealing to guide the transformation space searching process. To balance exploration and exploitation, they apply $\epsilon-$greedy in each searching iteration by selecting $\epsilon b$ candidates randomly to ensure exploration, where $b$ is the batch size and $\epsilon$ is a fixed value $0.05$. This decision appears suboptimal for several practical reasons. As shown in Fig. 4, a small $\epsilon$ (e.g., 0.01, 0.05) tends to be overly greedy, as it focuses in an area where the model believes the optimum to be, without efficiently exploring additional areas of the transformation space which may turn out to be more optimal in the long run. On the other hand, a large $\epsilon$ (e.g., 0.2, 0.5) induces a large distraction into the searching process and slows down the searching process. However, having a constant $\epsilon$, determined at the start of the beginning, introduces an additional hyperparameter that needs to be tuned. Furthermore, even with a tuned $\epsilon$, its value is going to remain the same during the entire optimization for all tasks, which is sub-optimal. The effectiveness of the existing approach is, therefore, significantly affected by the degree of trade-off between exploration and exploitation. Given that the general goal is to make the optimization fast and more usable, this is clearly not desirable. Thus, we need a more principled way to control the balance between exploration and exploitation.

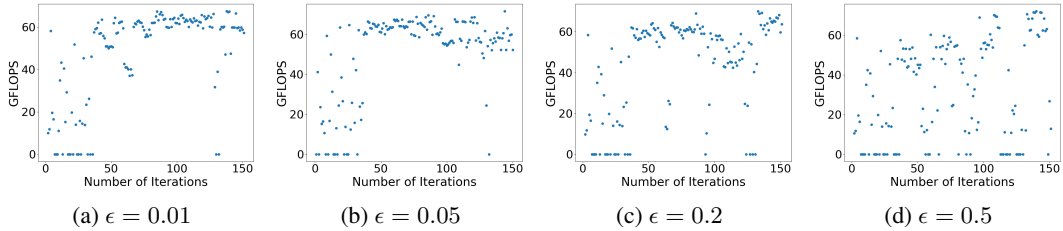

| (a) $\epsilon = 0.01$ | (b) $\epsilon = 0.05$ | (c) $\epsilon = 0.2$ | (d) $\epsilon = 0.5$ |

Figure 4: Exploration-vs-exploitation: choice of $\epsilon$ in $epsilon$-greedy.

# 4 The Elements of AdaTune

In this section, we present the design decision for AdaTune, and we provide quantified comparisons against corresponding configurations of the original AutoTVM [14] in Section 5. We illustrate the high-level design of AdaTune in Figure 5. There are three main contributions that AdaTune makes over the design choices of AutoTVM.

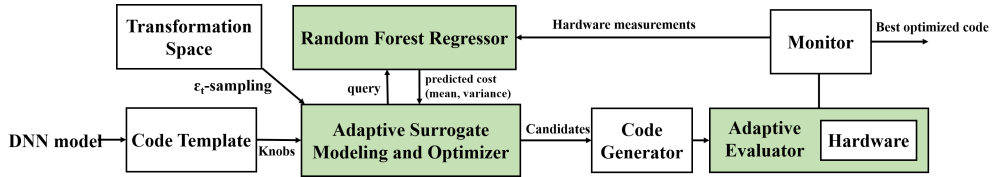

Figure 5: Overall design and optimization overview of AdaTune.

## 4.1 Adaptive Evaluator: Early Termination by Coefficient of Variation Counting

The adaptive evaluator (AE) is the module in charge of steering the dynamic reconfiguration process of batch measurement for getting the performance. The key challenge in the design of this component is to gather measurements in an accurate and timely way, so as to maximize accuracy (i.e., reduce variance) and minimize the cost for different hardware measurements.

In AdaTune, we use AE to automatically adjust the iterations of measurements $n$ in a model diversity and hardware heterogeneity aware way. This adaptive mechanism is based on the idea of estimating the statistical uncertainty associated with the current performance (e.g., GFLOPS) measurement on the basis of coefficient of variation (CV). More precisely, for a given $n$, we divide it into *micro-batches* of size $B$, and we evaluate the performance upon the finish of each micro-batch since the beginning of the hardware measurement (time = 0). If we denote $Time(i)$ ($i \in \{1, 2, ..., B\}$) as the time elapsed since the beginning of the measurement of plan $p$ and the occurrence of the i-th micro-batch, the GFLOPS upon the i-th micro-batch would be $Perf(p)_i = \frac{i \times GFLOP(op(p))}{Time(i)}$. We then use it to estimate the accuracy of the measurement after i micro-batches with the CV of $Perf(p)_i$, i.e., $\text{CV}(Perf(p)_i) = \frac{std(\{Perf(p)_1, Perf(p)_2, ..., Perf(p)_i\})}{avg(\{Perf(p)_1, Perf(p)_2, ..., Perf(p)_i\})}$. If the CV value is smaller than a certain threshold (e.g., 10%), AdaTune adaptively terminates a measurement. AE, therefore, automatically adjusts the hardware measurement costs in a robust and model/hardware-independent way.

Prior studies on hyperparameter tuning such as Hyperband [27] and BOHB [17] define approximate versions of the objective function (e.g., classification tasks) that are parameterized by a concept called budget. They prioritize promising configurations with larger budgets as the optimization progresses. Our approach is similar to the budget concept in the sense that we also create a cheap-to-evaluate version of the hardware measurement function. However, different from their goal, which is to eventually optimize with the largest budget, we collect a stream of measurements of micro-batches and timely stops when the likelihood of getting very different measurement results is low. [8] uses so-called adaptive sampling by performing non-uniform sampling through clustering. Different from their method, our approach cuts the cost of hardware measurement of individual samples. The two methods can be combined to maximize the gains.

## 4.2 Adaptive Surrogate Modeling and Optimizer

For AdaTune, we propose another two improvements: (1) We create a *surrogate model with uncertainty quantification*, which takes both mean and variance into consideration to adapt performance modeling and drives the exploration of the transformation space by continuously gathering feedback on the quality of the explored transformation plans; (2) We introduce the *contextual simulated annealing* optimizer, which dynamically balances the trade-off between exploration and exploitation based on the expected improvement from the surrogate model.

### 4.2.1 Surrogate modeling with uncertainty quantification

Given the very different variance behaviors (Section 3), in order to make the optimization process more adaptive to different hardware and models, we consider constructing a surrogate model by accounting for uncertainty. In particular, we consider an ensemble model $\tilde{f}$ of black-box learners, each of which is built on $m$ measurements randomly sampled with repetitions from the entire hardware measurements $\{(p_1, Perf(p_1)), ..., (p_m, Perf(p_m))\}$, where a transformation plan $p_i = (p_{i,1}, ..., p_{i,d})$ is a complete instantiation of the code template's $d$ knobs. Given a new plan $p_{m+1}$, the model predicts the mean $\mu$ and variance $\sigma$ for its performance $\tilde{f}(p_{m+1})$ through the ensemble.

Among many options, a useful tool for quantifying the uncertainty in a given prediction is random forest, which has proven to be valuable in Sequential Model-based Bayesian Optimization (SMBO) [11] method such as *SMAC* [23]. We choose random forest as our surrogate model because it enjoys advantages, such as better handling of discrete features. It also has a training time complexity of $O(m \cdot log(m) \cdot d \cdot H)$ where $H$ is the number of decision trees, which is more efficient than models like Gaussian Processes, which exhibit $O(m^3)$ training complexity in the number of data points (see comparison results in Appendix B).

**Expected positive improvement.** We use the same diversity-aware cost function as the one used in AutoTVM [14] to select a list of promising plans for hardware measurements. In particular, we replace the run time cost estimate part with *expected positive improvement* (EI) and keep the other term unchanged. We compute $EI(p) = \mathbb{E}[max(\tilde{f}(p) - Perf(p^*), 0)]$ [11] over the best known measured performance $Perf(p^*)$ so far ($p^*$ is the best plan so far), while taking into account the possible uncertainty in that prediction. Given the predictive mean $\mu$ and standard deviation $\sigma$ of a

plan $p$, we have:

$$EI(p) = \begin{cases} (\mu(\tilde{f}(p)) - Perf(p^*))\Phi(Z) + \sigma(\tilde{f}(p))\phi(Z) & if \; \sigma(\tilde{f}(p)) > 0 \\ max(0, \mu(\tilde{f}(p)) - Perf(p^*)) & if \; \sigma(\tilde{f}(p)) = 0 \end{cases} \quad (1)$$

$$Z = \begin{cases} \frac{\mu(\tilde{f}(p)) - Perf(p^*)}{\sigma(\tilde{f}(p))} & if \; \sigma(\tilde{f}(p)) > 0 \\ 0 & if \; \sigma(\tilde{f}(p)) = 0 \end{cases} \quad (2)$$

where $\Phi$ and $\phi$ are the CDF and PDF of the standard normal distribution. EI is possibly large for configurations with high predicted performance and for those with high predicted uncertainty.

#### 4.2.2 Adaptive control of exploration and exploitation via contextual simulated annealing

Based on the analysis in Section 3, we propose a modification to the $\epsilon$-greedy sampling in the simulated annealing based optimizer. Instead of using a fixed value for $\epsilon$, we replace $\epsilon$ with a contextual factor $\epsilon_t$, which is implicitly tied to the task and the underlying model and changes dynamically as the optimization proceeds. In particular, we define:

$$\epsilon_t = \frac{\bar{\sigma}}{Perf(p^*)} \quad (3)$$

where $Perf(p^*)$ is the best seen plan, and $\bar{\sigma}$ is the mean of the standard deviations from a set of yet unsampled plans from posterior distribution (rather than the prior). Note that it should be distinguished from $\sigma$, which is the individual standard deviation of a prediction from $\tilde{f}(\cdot)$ for a particular plan in the posterior. This allows AdaTune to dynamically adjust the exploration–exploitation trade-off based on the surrogate model's state at any time point. We call the resulting optimizer contextual simulated annealing.

This is intuitive, as exploration is, on average, preferred when the model has high uncertainty, and exploitation is preferred when the predicted uncertainty is low. Furthermore, if the optimization is being overly greedy (i.e., getting stuck at a local optimum), $\tilde{f}$ will produce a highly unbalanced standard deviation distribution with small standard deviation close to the local optimum already being sampled, and larger standard deviations elsewhere in the (unsampled) transformation space. This results in a larger value for the standard deviation for the posterior, which is equivalent to an increase of $\epsilon_t$ in Eqn. 3 and it increases the ratio of randomly sampled points in the next batch of hardware measurements, presumably helping the search escape from local optimum.

### 4.3 AdaTune: Putting It Together

In previous sections, we describe how we make the optimization process more adaptive and reduce the hardware measurement cost at each tuning step. In this part, we put everything together and call the resulting target-dependent optimization pass AdaTune (Algorithm 1).

## 5 Evaluation

In this section, we evaluate AdaTune experimentally, seeking answers to how AdaTune helps accelerate the optimization process. We integrate AdaTune with TVM [13] and use AutoTVM [14] as our baseline for comparison. We implement AdaTune in Python, and we leverage scikit-learn [33] and forestci [1] to implement the surrogate model and optimizer.

### 5.1 Comparison of AutoTVM and AdaTune for Searching Transformation Space

We compare the performance of AutoTVM and AdaTune on how much optimization speedup we obtain as a function of the wall-clock time. Due to space limitations, we include four tasks: one convolutional layer sampled from ResNet-18 [20] and one batched GEMM operator from Transformer [41] on both CPU (Intel Xeon CPU E5-2690 v3 @ 2.60GHz 2600 MHz) and GPUs (Nvidia Tesla P100). We use $n$=500 for all experiments and set micro-batch size $B = 50$ in AdaTune. We use the default settings for other hyperparameters provided by AutoTVM. The detailed parameter settings are included in Appendix A. We perform 15 independent runs of each configuration with different random seeds and report the median together with a 95% confidence interval. Also note that the predicted performance is only used in the transformation space searching process, and we report *real measured latency* in the end-to-end evaluation results.

**Algorithm 1**                                                                 **AdaTune**

1: **Input:** Transformation space $S_e$
2: **Output:** Selected transformation plan $p^*$
3: $D \leftarrow \{\}$
4: **while** $n\_iterations < max\_n\_iterations$ **do**
5:     Q $\leftarrow$ run contextual simulated annealing to collect candidates in $S_e$ using the surrogate model
    $\tilde{f}$ and EI in Section 4.2.1                               ▷ Finding the next promising batch
6:     Random sample $K$ plans $p_1, p_2, ..., p_K$ from $S_e$
7:     $\epsilon_t \leftarrow \frac{\frac{1}{K}\sum_{k=1}^{K}(standard\_deviation(\tilde{f}(p_k)))}{Perf(p^*)}$
8:     S $\leftarrow$ pick (1 - $\epsilon_t$)b subset from Q
9:     S $\leftarrow$ S $\cup$ {Randomly sample $\epsilon_t$b candidates}
10:     **for** p in S **do** do
11:         **for** i in (1,..,B) **do**                         ▷ Measure the hardware cost with AE
12:         $cv \leftarrow \frac{std(\{Perf(p)_1, Perf(p)_2, ..., Perf(p)_i\})}{avg(\{Perf(p)_1, Perf(p)_2, ..., Perf(p)_i\})}$
13:         **if** $cv < threshold$ **then**
14:             break
15:         $D \leftarrow D \cup (p, Perf(p))$
16:     update $\tilde{f}$ using D                       ▷ Update the model given new measurements
17:     $n\_iterations \leftarrow n\_iterations + b$
18: $p^* \leftarrow$ best found transformation plan

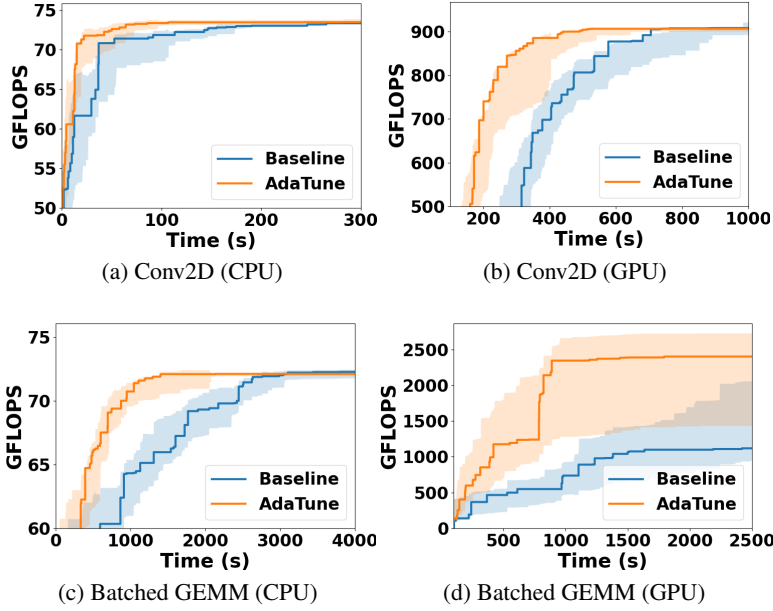

(a) Conv2D (CPU)                               (b) Conv2D (GPU)

(c) Batched GEMM (CPU)                     (d) Batched GEMM (GPU)

Figure 6: Comparison of optimization levels under the same time budget on both CPU and GPUs.

Fig 6 visualizes the results. The x-axis denotes the wall-clock time of auto-tuning. The y-axis denotes the GFLOPS of the best transformation plan found so far. We make the following observations. The first observation is that the best plan AdaTune finds is similar to, and sometimes much better than the baseline. The speedup is especially prominent when the transformation space is extremely large (e.g., Fig. 6d on GPUs), where AdaTune achieves up to 115% higher GFLOPS than the baseline under the same time budget. This indicates that AdaTune is able to explore the transformation space in a more efficient way. The second observation is that AdaTune is 1.3–3.9× faster than AutoTVM to find the best plan (Fig. 6a–6c). This improved speed to find the best plan is important for achieving better anytime performance in optimizing new models. These results confirm that AdaTune is capable of finding good code transformation at a much faster speed than the baseline.

Fig. 6d shows a much larger variance than the other figures because, for that workload, there are lots of zero points (invalid knobs) in the search space; thus the modeling is highly dependant on the nonzero points found at the beginning. If there is not enough exploration, the search tends to be trapped in local optima.

## 5.2 Comparison on Optimization Time and Model Performance Improvements.

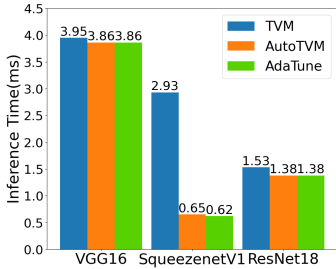
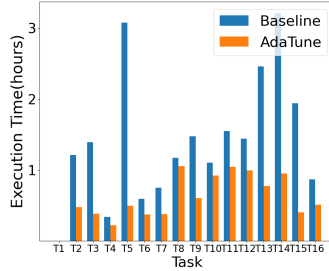
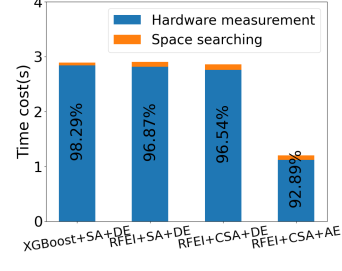

Figure 7: Inference time comparison.

Figure 8: Optimization wall-clock time comparison.

Figure 9: Breakdown comparison of tuning cost.

We compare the end-to-end optimization results on ResNet-18 [20], VGG16 [39], and Squeezenet-V1 [24]. Fig. 7 compares the final inference time from ResNet-18 optimized by TVM, AutoTVM, and AdaTune respectively. Overall, AdaTune achieves up to 4.6% faster inference speed over AutoTVM, and up to 78.8% faster speed over TVM, respectively. AutoTVM and AdaTune achieve much higher speedup on SqueezeNet, presumably because the heuristic-based optimizations in TVM are sub-optimal. In contrast, the Learning to Compile approach is able to quickly identify code transformation leads to a significantly faster speed. While achieving comparable optimization quality as AutoTVM, AdaTune significantly reduces the lengthy optimization time. Due to space limitations, Fig. 8 shows the optimization time on ResNet-18 on GPU. AutoTVM takes 22.6 hours in total for the optimization, whereas AdaTune takes only 9.6 hours to finish the optimization, which is a 2.35× speedup.

## 5.3 Ablation Analysis

In this section, we compare the effectiveness of design elements in AdaTune by comparing the following schemes:

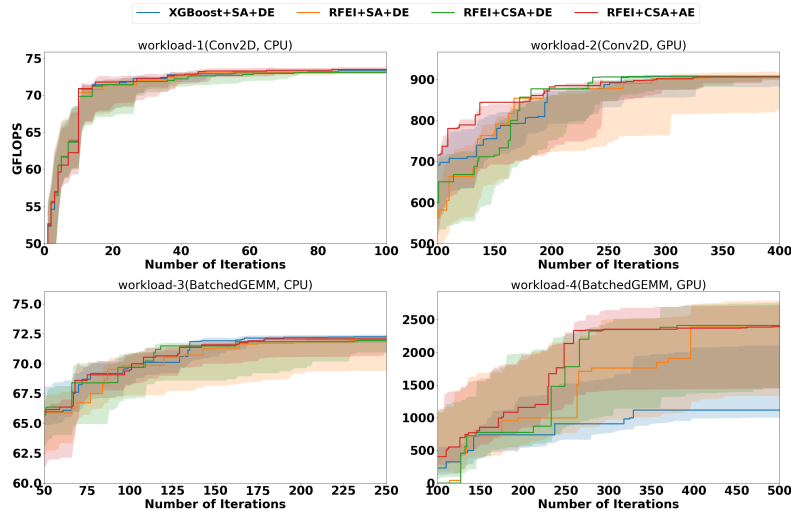

Figure 10: Impact to search effectiveness in iterations.

- XGBoost + SA + DE: This is our baseline, which uses XGBoost as the performance model, simulated annealing (SA) as the optimizer, and deterministic evaluator (DE).

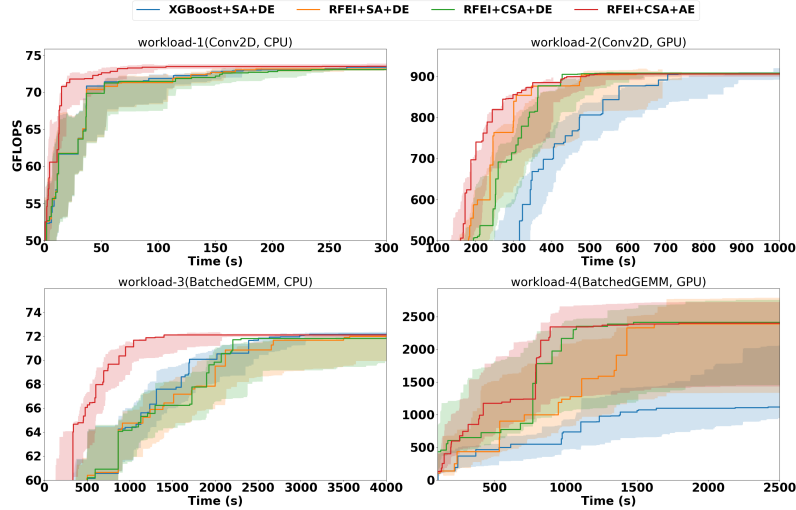

Figure 11: Impact to search effectiveness in wall-clock time.

- RFEI + SA + DE: Like the baseline, but XGBoost is replaced with RFEI. RFEI stands for random forest plus positive expected improvement.

- RFEI + CSA + DE: Like our approach but uses a deterministic evaluator instead of the adaptive evaluator. CSA stands for the contextual simulated annealing.

- RFEI + CSA + AE: Our main algorithm as described in Section 4.3. AE stands for the adaptive evaluator.

**Searching effectiveness.** The impact on the search effectiveness with respect to iterations is presented in Fig 10. When equipped with the uncertainty-quantifying surrogate model, the search takes a relatively smaller number of iterations to find a better plan (as shown in workload-4 from Fig. 10). In other cases, it performs similarly to the XGBoost model. Contextual simulated annealing further improves the searching effectiveness in some cases (workload 2 and 4 in Fig. 10), presumably because of its effect of regularizing the search to escape from local optima. Finally, with the adaptive evaluator, there is a significant improvement in wall-clock time on all the tasks, as shown in Fig 11.

**Cost breakdown.** Fig 9 further shows the breakdown in the average time required for transformation space searching (ResNet-18 on CPU) and hardware measurement in one iteration. The other components, such as program code generation, incur only a negligible amount of overhead. Overall, our surrogate model and contextual optimizer add very minimal overhead over the baseline. AdaTune effectively reduces the hardware measurement time by $2.5\times$, which contributes to the speedup of the end-to-end optimization time.

## 6   Conclusion

Although highly optimized code can be achieved through existing DL compilers, an obvious drawback is their long code optimization time, required to generate many versions of a tensor program and to profile these versions on hardware. In this paper we have introduced a method, called AdaTune, to make the code optimization process in DL compilers more adaptive to different hardware and models. The adaptive evaluator allows cut hardware measurement cost significantly without losing much accuracy. The uncertainty-aware surrogate model and the contextual optimizer allow us to more efficiently explore the transformation space. As a result, AdaTune achieves higher speedups in terms of finding a good transformation plan on different types of hardware and models, outperforming AutoTVM, a state-of-the-art approach.

## Broader Impact

Machine learning and deep learning applications are becoming ubiquitous in large scale production systems. With that growth and the scaling in model size and complexity, the focus on efficiently executing DNN models has become even greater. The push for increased energy efficiency has led to the emergence of diverse heterogeneous systems and hardware architectures. While it is possible to hire deployment engineers to produce highly optimized code for diverse architectures, such an approach is time-consuming. It requires significant manual effort, which is difficult to scale, as new DNN models and operators are coming out on a regular basis. Compilers have historically been the bridge between programming efficiency and high-performance code, which allows fast innovation while producing high-performance code for diverse architectures. Auto-tuning techniques such as AutoTVM modernize a compiler by automatically learning the compiler's optimization decisions as opposed to using heuristic rules. However, the actual cost of running such a tuning process is very expensive. Our techniques speed up the auto-tuning process significantly. It improves the agility of deploying DNN models, fostering fast innovations. It also reduces the amount of hardware resources needed for optimizing DNN models, reducing the corresponding energy consumption and carbon footprint produced.

## Acknowledgments and Disclosure of Funding

The authors are grateful for the discussion with Silu Huang, Eric Zhu, and Jon Soifer. The authors appreciate the anonymous NeurIPS reviewers for providing constructive feedback for improving the quality of this paper. All authors are not funded by any other agency.

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
