[Supplementary Material]

# A  Hyperparameter Settings

We treat all feature inputs as numeric inputs to the Random Forest model. we set the batch size to 32 (i.e., updating the cost model once with 32 new hardware measured points). For the Random Forest Regressor model, We set the number of trees to 10 to keep the computational overhead small. We set $max\_features$ to 10 to avoid over-fitting and use the default values for other settings in scikit-learn. When calculating the contextual $\epsilon$ value, we randomly sample 20 plans from the transformation space to obtain the prediction mean and variance.

# B  Additional Results

## B.1  Comparison with Gaussian Process

We compare two uncertainty estimators: Gaussian Process Regressor and Random Forest Regressor. Results show that Random Forest Regressor performs much better than Gaussian Process Regressor. Therefore, we choose the Random Forest Regressor as our cost model.

Figure 12: Random Forest vs. Gaussian Process performance.

## B.2  Additional Optimization Time Comparison Results

We include more results (Figure 2 and Figure 3) on the optimization time comparisons between AutoTVM and AdaTune. Overall, AdaTune achieves 1.3–2.9X speedup in end-to-end optimization time. The speedup on some GPU tasks is relatively lower because the hardware measurement on GPUs is faster than that of the same task on CPU.

|  | AutoTVM | AdaTune | Speedup |
|---|---|---|---|
| Resnet-18 | 22.6h | 9.6h | 2.4X |
| Resnet-50 | 20.0h | 14.1h | 1.4X |
| VGG-16 | 21.9h | 16.7h | 1.3X |
| SqueezenetV1 | 7.6h | 5.8h | 1.3X |
| Transformer (Enc.) | 3.8h | 2.8h | 1.4X |

Table 2: Optimization time on GPU.

|  | AutoTVM | AdaTune | Speedup |
|---|---|---|---|
| Resnet-18 | 2.0h | 1.0h | 2.0X |
| Resnet-50 | 3.6h | 1.7h | 2.1X |
| VGG-16 | 18.9h | 6.5h | 2.9X |
| SqueezenetV1 | 1.2h | 0.7h | 1.7X |
| Transformer (Enc.) | 8.4h | 3.8h | 2.2X |

Table 3: Optimization time on CPU.

## B.3  Additional Inference Time Comparison Results

We include more results (Figure 4 and Figure 5) on the inference time comparisons between AutoTVM and AdaTune. Although being faster to optimize, AdaTune achieves comparable optimization quality and sometimes outperforms AutoTVM in inference time.

## B.4  Additional Optimization Cost Results

We include detailed optimization time breakdown results for all tasks in ResNet-18, ResNet-50, VGG-16, Squeezenet-V1.1, and Encoder on both CPU and GPU (Figure 14 Figure 15, Figure 16, and Figure 17). Overall, AdaTune improves the optimization time for individual tasks on both CPU

|  | TVM | AutoTVM | AdaTune |
|---|---|---|---|
| Resnet-18 | 1.53ms | 1.38ms | 1.38ms |
| Resnet-50 | 4.82ms | 4.37ms | 4.37ms |
| VGG-16 | 3.95ms | 3.86ms | 3.86ms |
| SqueezenetV1 | 2.93ms | 0.65ms | 0.63ms |
| Transformer (Enc.) | 78.15ms | 52.25ms | 47.46ms |

Table 4: Inference time comparison on GPU.

|  | TVM | AutoTVM | AdaTune |
|---|---|---|---|
| Resnet-18 | 79.24ms | 52.64ms | 52.64ms |
| Resnet-50 | 217.12ms | 115.76ms | 115.68ms |
| VGG-16 | 884.94ms | 442.01ms | 438.68ms |
| SqueezenetV1 | 14.41ms | 11.36ms | 11.25ms |
| Transformer (Enc.) | 2897.27ms | 1620.88ms | 1607.67ms |

Table 5: Inference time comparison on CPU.

and GPU for the models being tested. On GPU, sometimes AdaTune takes a slightly longer time in certain tasks (e.g., T18 in VGG-16 and T1 in SqueezeNet-V1.1). That is because the auto-tuning process stops when it can no longer find a better solution. We find that AutoTVM sometimes stops earlier because it quickly gets stuck at a local optimum. In contrast, the contextual optimizer in AdaTune constantly pushes AdaTune out of local optimum, which yields a longer time for AdaTune to trigger the stop condition.

(a) ResNet-18, GPU      (b) ResNet-18, CPU

Figure 13: Optimization wall-clock time comparison for ResNet-18.

(a) ResNet-50, GPU      (b) ResNet-50, CPU

Figure 14: Optimization wall-clock time comparison for ResNet-50.

(a) VGG-16, GPU      (b) VGG-16, CPU

Figure 15: Optimization wall-clock time comparison for VGG-16.

(a) SqueezenetV1.1, GPU      (b) SqueezenetV1.1, CPU

Figure 16: Optimization wall-clock time comparison for SqueezenetV1.1.

(a) Transformer (Enc.), GPU  (b) Transformer (Enc.), CPU

Figure 17: Optimization wall-clock time comparison for Encoder.