[Reviews · NeurIPS 2020]

Review 1

Summary and Contributions: The authors present a new autotuning algorithm with the following improvements. * early termination of evaluation runs if the CV of measurements is less than a threshold * adaptive surrogate modeling and using it to adaptively control exploitation vs exploration factor - This biases the search algorithm (in this case simulated annealing) to explore more if the model prediction is uncertain and to exploit more when the performance prediction is more certain.

Strengths: * The authors correctly identify the need for adaptive techniques in program auto-tuning. They demonstrate this in 3 contexts, adaptive early termination of evaluation runs, adaptive selection of candidates using a surrogate model, adaptive exploration vs exploitation threshold (this uses the surrogate model). * I really liked section 3 with observations. It gives insights as to why adaptive techniques for program auto-tuning is needed. * Their evaluation on the selected benchmarks compared to AutoTVM.

Weaknesses: OpenTuner * OpenTuner (http://opentuner.org) is a general-purpose auto-tuning framework that is widely used in compiler / automatic program optimization. The authors do not mention or compare against them. OpenTuner is built for tuning various knobs in a compiler / automatic program optimization setting. * In OpenTuner, the authors use a list of search techniques and then use an AUC meta technique to select the best search technique to use at a given time. The authors only use simulated annealing. I would like to see how the optimization times compare with OpenTuner which can adaptively use different search techniques. * I feel an evaluation with OpenTuner is essential to understand the state-of-the-art in program auto-tuning. Evaluation on larger NNs * Results from optimizing ResNet-50 would be more compelling. * End-to-end evaluation on a transformer model would also give evidence that this technique scales pretty well (may be BERT). Expected Improvement * It is not clear from the text how EI is used to select a promising plan. It would be better to mention how this is used in the fitness function in simulated annealing (line 5). Is it the entire fitness function or part of it? ****** post rebuttal comments ************ Thank you for the rebuttal response. Overall, the adaptive techniques introduced in this paper are valuable to the community. However, I would suggest the following changes to make the paper complete and correct w.r.t. to other works. * Since, the publication of the original AutoTVM paper, a few works that improve on this baseline have been published (including Adams et. al.). Therefore, I would urge the authors to change their state-of-the-art performance claims to "state-of-the-art with respect to a simulated annealing baseline". * Please add a citation to OpenTuner as well as discuss it in the paper (summary of the rebuttal response would be good due to page constraints).

Correctness: I believe that the empirical results shown in this paper are correct.

Clarity: Overall, the paper is well-written and easy to follow. Section 4.2.1 can be written better with more intuition.

Relation to Prior Work: * OpenTuner is not discussed nor compared against. It is a widely used compiler auto-tuning framework and is a key related prior work. I find this omission significant.

Reproducibility: Yes

Additional Feedback: * Please address my major concerns mentioned under weaknesses. The following are few minor details. * Figure 11 and 12, shouldn't the orange line climb up to the peak GFLOP plan faster than the rest to support your conclusions in section 5.3? Please explain if I am reading the graph wrong. * line 5 in algorithm 1 needs more explanation.


Review 2

Summary and Contributions: This paper proposes several methods to make adaptive tensor program compilation efficient. The contribution includes: - An analysis that reveals the inefficiency and challenges of the existing approaches - Several methods to make adaptive optimization more efficient

Strengths: - Insightful analysis of the inefficiency in existing approaches - Good and consistent results across different hardware platforms - Detail evaluation and ablation study

Weaknesses: - The evaluation section is confusing and not well-organized

Correctness: The claim and methods in the paper are correct and follow the widely accepted practice.

Clarity: This paper provides enough background information, but the evaluation section is not well organized and makes me confused. - Sec 5.2 Fig.7 looks good to me, but I think it is just a microbenchmark because it only tests four small cases. Then Fig.8, Fig.9, and Fig.10 confuse me a lot. Fig.9 is on GPU but Fig.10 is on CPU. Whether Fig.8 is on GPU or CPU is not documented. I would like to see all these comparisons on both CPU and GPU. - Sec 5.3 The main algorithm (RFEI + CSA + AE) in Fig. 12 performs worse than (RFEI + CSA + DE) on all cases in FIg.12. Why does that happen? Does it mean AE is useless at all?

Relation to Prior Work: This paper clearly discusses the its difference from previous works.

Reproducibility: Yes

Additional Feedback: The overall idea of this paper sounds good. But the evaluation section needs more work to make it clear. Typo: L233: Delete the redundant "2600 MHz" ========== Post Rebuttal Comments ========== Thanks for the response. It clarifies my concerns about the evaluation section, which are accidentally caused by the typo in Fig. 11 and Fig. 12. I would like to raise my score to 7.


Review 3

Summary and Contributions: The authors present an adaptive tensor compilation framework that vastly reduces the time needed to optimize tensor programs (e.g. ML pipelines) while matching or exceeding the performance of state of the art approaches.

Strengths: The performance of neuronal networks is crucial, especially since inference and training can consume vast amounts of compute. Optimizing these programs to run faster and be more efficient is a crucial aspect of ML research and highly relevant to the NeuroIPS audience. The authors demonstrate a significant reduction of time to optimize these models, while maintaining or exceeding the performance of the current state of the art.

Weaknesses: No major weaknesses

Correctness: Yes

Clarity: Very well written

Relation to Prior Work: Yes it relation to prior work is clearly established

Reproducibility: Yes

Additional Feedback: Thank you for writing such a clear and lucid paper, I haven't had the opportunity before to look at learn to compile, but your paper spawned an interest to learn more about the topic.

[Author Response · NeurIPS 2020]

*Answers to Reviewer #1*.

Q1: Comparison to OpenTuner.

A: We thank the reviewer for pointing out the OpenTuner project. OpenTuner identifies the importance of having
domain-specific search techniques in auto-tuning. It was built to adaptively choose among domain-specific search
techniques for general program tuning. In contrast, our focus is on NN model compilation rather than generic program
tuning. Even though OpenTuner may be regarded as the state-of-the-art for other problems, AutoTVM is more widely
accepted as the state-of-the-art in NN model compilation. Our technique is designed and validated specifically for the
NN model compilation problem, and we are not claiming we advance the generic problem of program auto-tuning.
Furthermore, our technical contribution is beyond the specific search strategies employed during auto-tuning, such as
those used in OpenTuner. There can be multiple paths to advance the state-of-the-art of NN model compilation, and
our work focuses on one of them. Finally, we find that OpenTuner is not directly applicable to our problem without
non-trivial modifications. OpenTuner does not support optimizing PyTorch or TensorFlow models out-of-the-box and
requires the user to manually create a configuration manipulator for the target program before optimization. However, it
is non-trivial to create such a configuration manipulator for neural network components as their implementation details
(e.g., tiling decisions) are often hidden away from end users. Even if we can create such a configuration manipulator,
doing so for all tunable operators in a neural network seems time-consuming and tedious because there can be tens or
hundreds of them. In contrast, AdaTune is closely coupled with AutoTVM, which automatically generates tunable code
templates for neural network components. We will add a citation describing OpenTuner.

Q2: Evaluation on larger NNs on Transformer and ResNet-50.

A: We added additional experiments on Transformer and ResNet-50 in the appendix (also shown results below).
AdaTune is 1.4-2.2X faster in optimization time while achieving comparable and sometimes better inference time
than the baseline. We also observe that larger models do not necessarily indicate longer optimization time. Take the
Transformer as an example. Since each layer of the model has the same model structure, AdaTune only needs to
optimize it once and apply the same optimization strategy across all Transformer layers to reduce its latency.

|                   | AutoTVM | AdaTune | Speedup |
|-------------------|---------|---------|---------|
| Resnet-18         | 22.6h   | 9.6h    | 2.4X    |
| Resnet-50         | 20.0h   | 14.1h   | 1.4X    |
| VGG-16            | 21.9h   | 16.7h   | 1.3X    |
| SqueezenetV1      | 7.6h    | 5.8h    | 1.3X    |
| Transformer (Enc.)| 3.8h    | 2.8h    | 1.4X    |

Table 1: Optimization time on GPU.

|                   | AutoTVM | AdaTune | Speedup |
|-------------------|---------|---------|---------|
| Resnet-18         | 2.0h    | 1.0h    | 2.0X    |
| Resnet-50         | 3.6h    | 1.7h    | 2.1X    |
| VGG-16            | 18.9h   | 6.5h    | 2.9X    |
| SqueezenetV1      | 1.2h    | 0.7h    | 1.7X    |
| Transformer (Enc.)| 8.4h    | 3.8h    | 2.2X    |

Table 2: Optimization time on CPU.

|                   | TVM     | AutoTVM | AdaTune |
|-------------------|---------|---------|---------|
| Resnet-18         | 1.53ms  | 1.38ms  | 1.38ms  |
| Resnet-50         | 4.82ms  | 4.37ms  | 4.37ms  |
| VGG-16            | 3.95ms  | 3.86ms  | 3.86ms  |
| SqueezenetV1      | 2.93ms  | 0.65ms  | 0.63ms  |
| Transformer (Enc.)| 78.15ms | 52.25ms | 47.46ms |

Table 3: Inference time comparison on GPU.

|                   | TVM       | AutoTVM   | AdaTune   |
|-------------------|-----------|-----------|-----------|
| Resnet-18         | 79.24ms   | 52.64ms   | 52.64ms   |
| Resnet-50         | 217.12ms  | 115.76ms  | 115.68ms  |
| VGG-16            | 884.94ms  | 442.01ms  | 438.68ms  |
| SqueezenetV1      | 14.41ms   | 11.36ms   | 11.25ms   |
| Transformer (Enc.)| 2897.27ms | 1620.88ms | 1607.67ms |

Table 4: Inference time comparison on CPU.

Q3: "It is not clear from the text how EI is used to select a promising plan. Is it the entire fitness function or part of it?"

A: We use the same diversity-aware function as the one used in AutoTVM, which is the addition of two terms: one is
for the runtime cost estimate and the other considers the diversity when selecting candidates. We replace the run time
cost part with EI and keep the second term unchanged to have a fair comparison.

Q4: "Figure 11 and 12, shouldn't the orange line climb up to the peak GFLOP plan faster?"

A: Thanks for pointing out. Since orange and red are similar colors, we accidentally switched the RFEI+CSA+AE and
RFEI+CSA+DE label when generating the shared legend as a separate picture. We will use more distinguishable colors.

*Answers to Reviewer #2*.

Q1: "Whether Fig.8 is on GPU or CPU. I would like to see all these comparisons on both CPU and GPU."

A: Fig.8's result is on GPU. In the appendix (Table 1-4 and Figure 14-17), we included optimization time and inference
time on both CPU and GPU. For the hardware measurement vs. space searching cost ratio on GPU (ResNet-18), it is
28% : 72% (22.6h) for the baseline and 44% : 56% (9.6h) for AdaTune. AdaTune reduces the hardware measurement
time by 1.5X and reduces the space searching cost by 3X. Together it speedups the optimization by 2.4X.

*Answers to Reviewer #3*.

We thank the reviewer for the positive feedback and for highlighting the significance of our work.

[Meta-Review · NeurIPS 2020]

This paper was discussed by reviewers after reading the authors’ feedback. Reviewers agree that the adaptive techniques introduced in this paper are valuable to the community. Therefore this paper is accepted. Please do update the paper according to reviewers’ feedback(in particular the point brought up by R1 in the updated review).